

# The pretreatment albumin to globulin ratio predicts survival in patients with natural killer/T-cell lymphoma

Xi-wen Bi[1,2,3], Liang Wang[2,3,4], Wen-wen Zhang[2,3,5], Shu-mei Yan[2,3,6], Peng Sun[1,2,3], Yi Xia[1,2,3], Zhi-ming Li[1,2,3] and Wen-qi Jiang[1,2,3]

[1] Department of Medical Oncology, Sun Yat-sen University Cancer Center, Guangzhou, Guangdong, PR China
[2] State Key Laboratory of Oncology in South China, Guangzhou, Guangdong, PR China
[3] Collaborative Innovation Center for Cancer Medicine, Guangzhou, Guangdong, PR China
[4] Department of Hematologic Oncology, Sun Yat-sen University Cancer Center, Guangzhou, Guangdong, PR China
[5] Department of Radiation Oncology, Sun Yat-sen University Cancer Center, Guangzhou, Guangdong, PR China
[6] Department of Pathology, Sun Yat-sen University Cancer Center, Guangzhou, Guangdong, PR China

Corresponding author
Wen-qi Jiang, wenqi_jiang@163.com

## ABSTRACT

**Background.** The pretreatment albumin to globulin ratio (AGR) has been reported to be a predictor of survival in several types of cancer. The aim of this study was to evaluate the prognostic impact of AGR in patients with natural killer/T-cell lymphoma (NKTCL).

**Methods.** We retrospectively reviewed the available serum biochemistry results for 331 NKTCL patients before treatment. AGR was calculated as albumin/(total protein—albumin), and a cut-off value of 1.3 was used to define AGR as low or high. Survival analysis was used to assess the prognostic value of AGR.

**Results.** A low AGR (<1.3) was associated with significantly more adverse clinical features, including old age, poor performance status, advanced stage, elevated lactate dehydrogenase, B symptoms, and high International Prognostic Index (IPI) and natural killer/T-cell lymphoma prognostic index (NKPI) scores. Patients with a low AGR had a significantly lower 5-year overall survival (44.5 vs. 65.2%, $P < 0.001$) and progression-free survival (33.1 vs. 57.4%, $P < 0.001$). In the multivariate analysis, a low AGR remained an independent predictor of poorer survival. Additionally, AGR distinguished patients with different outcomes in the IPI low-risk group and in the NKPI high-risk group.

**Discussion.** Pretreatment AGR may serve as a simple and effective predictor of prognosis in patients with NKTCL.

## INTRODUCTION

Extranodal natural killer/T-cell lymphoma (NKTCL) is a relatively rare lymphoid malignancy (*Chim et al., 2004*; *Vose, Armitage & Weisenburger, 2008*). Most cases of NKTCL initially arise in the nasal cavity and are diagnosed at an early stage. However, destructive and ulcerative lesions and the frequent involvement of neighboring tissues indicate an invasive biological behavior (*Chan, Jaffe & Ralfkiaer, 2001*; *Chan et al., 2008*). NKTCL is highly radiosensitive, and radiotherapy (RT) has been well established as the primary treatment for localized disease (*Li et al., 2006*; *Li et al., 2012*; *Bi et al., 2013*). Probably due to the expression of a multidrug-resistant gene in tumor cells, NKTCL is relatively refractory to anthracycline-based chemotherapy regimens (*Wang et al., 2008*; *Kim et al., 2009*). However, novel regimens containing L-asparaginase (L-ASP) or pegaspargase have shown encouraging results (*Jaccard et al., 2011*; *Yamaguchi et al., 2011*; *Kwong et al., 2012*). A heterogeneous treatment response and prognosis have been observed in patients with NKTCL, making accurate prognostic stratification at diagnosis important. Previous studies have identified various prognostic factors in NKTCL, such as the stage of disease, local tumor invasion, B symptoms, and the plasma Epstein-Barr virus (EBV) DNA copy number (*Lee et al., 2006*; *Au et al., 2009*; *Wang et al., 2012*; *Bi et al., 2015b*). The International Prognostic Index (IPI) and natural killer/T-cell lymphoma prognostic index (NKPI) have been the most commonly used models for prognostic stratification in NKTCL (*Shipp et al., 1993*; *Chim et al., 2004*; *Lee et al., 2006*). However, the value of currently reported prognostic indicators of NKTCL remains controversial because: (1) the prognostic power of some factors could not be consistently verified among different cohorts, such as local tumor invasion and IPI score (*Cheung et al., 2002*; *Lee et al., 2006*); and (2) distribution of patients into different risk groups based on the currently used IPI or NKPI models was disproportional with a majority of patients in the low-risk group, as previously reported (*Chim et al., 2004*; *Huang et al., 2011*).

It is well accepted that inflammatory reactions play an important role in promoting tumor progression and invasion, reducing the treatment response and worsening the prognosis (*Colotta et al., 2009*). Hypoalbuminemia is usually related to malnutrition. However, previous studies have found that hypoalbuminemia may be used as an effective indicator of up-regulated cancer-related inflammatory response, which is mainly attributed to the cytokine-induced suppression of albumin synthesis and increased albumin degradation (*Andersson et al., 1990*; *Fearon et al., 1998*). Serum globulins play a key role in immunity and inflammation, and elevated levels of serum globulin are usually associated with enhanced inflammatory response (*Gabay & Kushner, 1999*). Both decreased albumin levels and increased globulin levels have been associated with up-regulated inflammation and poor outcome for several types of cancer (*Boonpipattanapong & Chewatanakornkul, 2006*; *Siddiqui et al., 2007*; *Gabay & Kushner, 1999*; *Gupta & Lis, 2010*; *Guthrie et al., 2013*; *McMillan et al., 2001*). Therefore, a low albumin to globulin ratio (AGR) may amplify the predictive power of these two elements and serve as a more robust predictor of an enhanced inflammatory and adverse survival in cancer patients (*Azab et al., 2013a*; *Azab et al., 2013b*). In recent studies, low AGR has been reported to be an adverse prognostic factor

in patients with colorectal cancer (*Azab et al., 2013b*; *Shibutani et al., 2015*), breast cancer (*Azab et al., 2013a*), and lung cancer (*Duran et al., 2014*). A strong association between NKTCL and inflammatory reactions has been proposed. Pathologically, the lesions are characterized by coagulative necrosis and a diffuse inflammatory infiltrate. Clinically, ulcerative and destructive lesions, purulent rhinorrhea, and fever are very common manifestations (*Chan, Jaffe & Ralfkiaer, 2001*; *Chan et al., 2008*). However, previously identified prognostic factors of NKTCL were mainly related to tumor burden and patient status, and rarely focused on lymphoma-related inflammatory markers. Therefore, as an effective indicator of cancer related inflammatory reactions, the prognostic value of AGR may be worth further study in NKTCL. In this study, we retrospectively collected data related to AGR in a large cohort of patients with NKTCL and analyzed correlations with clinical features and prognosis.

## MATERIALS AND METHODS

### Patient selection and baseline evaluation

The medical records of 385 previously untreated patients with NKTCL who were treated at Sun Yat-sen University Cancer Center between 2001 and 2013 were reviewed. The inclusion criteria included: (1) diagnosis of NKTCL according to the World Health Organization (WHO) classification of lymphomas (*Chan, Jaffe & Ralfkiaer, 2001*; *Chan et al., 2008*); (2) complete laboratory data before initial treatment; and (3) complete follow-up data. The exclusion criteria were: (1) known active inflammatory disorders including autoimmune disease and infection; (2) known active liver or kidney disease; and (3) receiving supportive care alone, without chemotherapy or radiotherapy (RT). A total of 331 patients were included in this study. Written informed consent for the collection of medical information was obtained from all patients at the first visit. The ethics committee of Sun Yat-sen University Cancer Center approved this study (No. B2015-054-01).

Baseline evaluations included a medical history and physical examination; complete blood count; serum biochemistry (including lactate dehydrogenase (LDH) level); computed tomography and/or magnetic resonance imaging of the head and neck; computed tomography of the chest, abdomen, and pelvis; and a bone marrow examination. Patients were staged according to the Ann Arbor system. The International Prognostic Index (IPI, including age, performance status, stage of disease, LDH level, and number of extranodal lesions) and the natural killer/T-cell lymphoma prognostic index (NKPI, including stage of disease, the involvement of regional lymph nodes, B symptoms, and LDH level) were calculated for all patients (*Shipp et al., 1993*; *Lee et al., 2006*). The primary sites of disease were classified into the upper aerodigestive tract (UAT, including the nasal cavity, Waldeyer's ring, hypopharynx, larynx, and oral cavity) and the extra-UAT (any site other than a UAT site) (*Kim et al., 2008*). Total serum protein and serum albumin were measured using an automated immunoturbidimetric analyzer (7600-020; Hitachi High-Technologies, Tokyo, Japan) within seven days before initial treatment. The AGR was calculated as albumin/(total protein−albumin).

## Treatment

The treatment modality has been described previously (*Bi et al., 2015a*). Patients with early stage disease received induction chemotherapy followed by involved-field radiotherapy (IFRT). Patients with advanced disease received chemotherapy as the primary treatment, and IFRT could be delivered as consolidation, palliative or salvage therapy according to the physician's clinical judgment. As previously reported (*Bi et al., 2015a*; *Bi et al., 2015b*), chemotherapy regimens varied over the study period but were categorized as an asparaginase (ASP)-containing or anthracycline-based regimen, depending on whether L-ASP or pegaspargase was incorporated. IFRT was administered in daily fractions of 2.0–2.5 Gy (5 days a week), with a median dose of 54.6 Gy (range, 18.0–74.0 Gy).

## Follow-up and statistical analysis

The follow-up schedule has been reported previously (*Wang et al., 2014*). The overall survival (OS) was measured from the date of diagnosis to the date of death due to any cause or the date of the most recent follow-up. Progression-free survival (PFS) was measured from the date of diagnosis to the date of disease progression, relapse, death due to any cause, or the most recent follow-up. The survival data were calculated using the Kaplan–Meier method and compared using the log-rank test. Continuous variables were presented as medians (range) and were compared using the Mann–Whitney $U$-test. Categorical variables were reported as frequencies and percentages and were compared using the Chi-squared test. The Cox proportional hazard model was used for a univariate screen of all potential predictors of survival. Variables with statistical and clinical significance were included in the multivariate analysis using the stepwise forward Cox regression model. Results were considered statistically significant with a two-sided $P$ value $< 0.05$. The statistical analysis was performed using SPSS version 17.0 software (SPSS A, Inc., Chicago, IL, USA).

# RESULTS

## Determination of the optimal cut-off value for AGR

The optimal cut-off point to define a high or low AGR was determined using the method proposed by *Igarashi et al. (2001)*. Using the log-rank test, we examined the discriminative power of different cut-off values for the prediction of OS from 1.0 to 2.0 in steps of 0.1. The cut-off value at which the OS curves separated most significantly (i.e., 1.3) was selected to categorize the AGR as low or high (Table 1). A total of 117 (35.3%) patients had a low AGR and 214 (64.7%) had a high AGR.

## Clinical features and treatment

In Table 2, the clinical characteristics and treatment modalities of the cohort are presented and compared between patients with a low or high AGR. The cohort primarily comprised males (male/female ratio, 2.2:1). The median age of the patients was 42 years, with 15.4% of patients aged older than 60 years. Most of the patients presented with good performance status (84.3%), primary disease originating from UAT (88.8%), and early-stage disease (81.3%). A total of 52.3 and 29.0% of the patients had B symptoms and elevated LDH levels, respectively. The IPI was scored as 0–1 in 78.2% of patients, and the NKPI score was 0–2 in 81.0% of patients.

**Table 1** Differentiating power of cut-off values for pretreatment albumin to globulin ratio on overall survival of patients with NK/T-cell lymphoma.

| Cut-off value | No. of patients (low/high) | Overall survival | |
|---|---|---|---|
| | | Chi-squared | *P* value |
| 1.0 | 24/307 | 5.174 | 0.023 |
| 1.1 | 52/279 | 6.620 | 0.010 |
| 1.2 | 80/251 | 9.589 | 0.002 |
| 1.3 | 117/214 | 17.418 | 0.000 |
| 1.4 | 172/159 | 6.662 | 0.010 |
| 1.5 | 220/111 | 3.206 | 0.073 |
| 1.6 | 258/73 | 2.563 | 0.109 |
| 1.7 | 289/42 | 2.618 | 0.106 |
| 1.8 | 304/27 | 1.953 | 0.162 |
| 1.9 | 314/17 | 3.914 | 0.048 |
| 2.0 | 321/10 | 3.583 | 0.058 |

The baseline characteristics differed significantly between patients with a low or high AGR. As shown in Table 2, patients in the low AGR group presented with significantly more adverse clinical features, including advanced disease (stage III-IV: 25.6 vs. 15.0%, $P = 0.017$), elevated LDH (40.2 vs. 22.9%, $P = 0.001$), B symptoms (73.5 vs. 40.7%, $P < 0.001$), involvement of regional lymph nodes (49.6 vs. 28.5%, $P < 0.001$), and a higher IPI score ($P < 0.001$) and NKPI score ($P < 0.001$). Patients with a low AGR were slightly older (>60 years: 20.5 vs. 12.6%, $P = 0.057$), had a significantly worse Eastern Cooperative Oncology Group (ECOG) performance score ($\geq 2$: 28.2 vs. 8.9%, $P < 0.001$), and had a lower median body mass index (BMI, 20.4 vs. 21.4, $P = 0.010$). Serum albumin (median: 38.0 vs. 43.5, $P < 0.001$) and absolute lymphocyte count (median: 1.4 vs. 1.6, $P = 0.032$) were significantly lower in the low AGR group compared with the high AGR group, while C-reactive protein (CRP; median: 12.8 vs. 3.9, $P < 0.001$) and the erythrocyte sedimentation rate (ESR, 41 vs. 11, $P < 0.001$) were markedly higher in the low AGR group. In addition, primary lesions in the upper aerodigestive tract were less frequently observed in the low AGR group (83.8 vs. 91.6%, $P = 0.031$). Serum creatinine was higher in the high AGR group than compared with the low AGR group (median: 69.3 vs. 64.1, $P = 0.011$).

As shown in Table 2, 67.7% of patients underwent RT with or without chemotherapy as the first-line treatment, while 32.3% received chemotherapy alone. Anthracycline-based and ASP-containing chemotherapeutic regimens were administered to 60.7 and 39.3% of patients, respectively. The median dose of radiation was 54.6 Gy (range, 18.0–74.0 Gy), and the median number of chemotherapy cycles was 4 (range, 0–10). The treatment modalities differed significantly between patients in the low and high AGR groups (Table 2). There were significantly more patients receiving chemotherapy alone without RT in the low AGR group than in the high AGR group (43.6 vs. 26.2%, $P = 0.001$). ASP-containing chemotherapeutic regimens were used more frequently in the low AGR group than in the high AGR group (49.1 vs. 34.0%, $P = 0.008$). However, the dose of radiation and number of chemotherapy cycles were comparable between the two groups.

**Table 2  The clinical characteristics and treatment modalities of patients with NK/T-cell lymphoma.**

| Parameters[a] | Total n (%) | AGR < 1.3 n (%) | AGR ≥ 1.3 n (%) | P value |
|---|---|---|---|---|
| Overall | 331 (100) | 117 (100) | 214 (100) | |
| Male gender | 227 (68.6) | 75 (64.1) | 152 (71.0) | 0.194 |
| Age (years) | 42 (11–80) | 44 (11–77) | 41 (13–80) | 0.069 |
| Age > 60 years | 51 (15.4) | 24 (20.5) | 27 (12.6) | 0.057 |
| BMI (kg/m$^2$) | 21.2 (13.7–44.0) | 20.4 (13.7–44.0) | 21.4 (14.8–44.0) | 0.010 |
| BMI < 18.5 kg/m$^2$ | 53 (16.0) | 26 (22.2) | 27 (12.6) | 0.023 |
| ECOG score ≥ 2 | 52 (15.7) | 33 (28.2) | 19 (8.9) | 0.000 |
| Primary site | | | | |
| UAT | 294 (88.8) | 98 (83.8) | 196 (91.6) | 0.031 |
| EUAT | 37 (11.2) | 19 (16.2) | 18 (8.4) | |
| Ann Arbor stage | | | | |
| I–II | 269 (81.3) | 87 (74.4) | 182 (85.0) | 0.017 |
| III–IV | 62 (18.7) | 30 (25.6) | 32 (15.0) | |
| B symptoms | 173 (52.3) | 86 (73.5) | 87 (40.7) | 0.000 |
| Elevated LDH | 96 (29.0) | 47 (40.2) | 49 (22.9) | 0.001 |
| Involvement of regional lymph nodes | 119 (36.0) | 58 (49.6) | 61 (28.5) | 0.000 |
| Extranodal sites ≥ 2 | 44 (13.3) | 21 (17.9) | 23 (10.7) | 0.065 |
| IPI score | | | | |
| Low risk (0–1) | 259 (78.2) | 61 (69.2) | 178 (83.2) | 0.000 |
| Intermediate risk (2–3) | 50 (15.1) | 20 (17.1) | 30 (14.0) | |
| High risk (4–5) | 22 (6.6) | 16 (13.7) | 6 (2.8) | |
| NKPI score | | | | |
| Low risk(0) | 101 (30.5) | 12 (10.3) | 89 (41.6) | 0.000 |
| Intermediate risk(1–2) | 167 (50.5) | 68 (58.1) | 99 (46.3) | |
| High risk (3–4) | 63 (19.0) | 37 (31.6) | 26 (12.1) | |
| Total protein (g/L) | 72.1 (32.5–89.2) | 72.2 (32.5–89.2) | 72.0 (41.5–85.7) | 0.450 |
| Serum albumin (g/L) | 41.5 (21.6–54.4) | 38.0 (21.6–45.6) | 43.5 (26.8–54.4) | 0.000 |
| Hypoalbuminemia (<35 g/L) | 38 (11.5) | 29 (24.8) | 9 (4.2) | 0.000 |
| WBC count (k/cc) | 5.9 (0.9–23.0) | 5.7 (1.6–15.6) | 6.0 (0.9–23.0) | 0.262 |
| Neutrophil count (k/cc) | 3.5 (0.4–20.1) | 3.3 (0.6–12.9) | 3.5 (0.4–20.1) | 0.644 |
| Lymphocyte count (k/cc) | 1.5 (0.2–5.4) | 1.4 (0.2–4.5) | 1.6 (0.3–5.4) | 0.032 |
| CRP ($n = 246$, mg/L) | 6.6 (0.2–154.9) | 12.8 (0.5–154.9) | 3.9 (0.2–87.6) | 0.000 |
| ESR ($n = 117$, mm/h) | 17 (1–110) | 41 (3–104) | 11 (1–110) | 0.000 |
| Serum creatinine (μmol/L) | 67.1 (31.6–116.0) | 64.1 (32.8–108.0) | 69.3 (31.6–116.0) | 0.011 |
| Treatment modalities | | | | |
| Chemotherapy alone | 107 (32.3) | 51 (43.6) | 56 (26.2) | 0.001 |
| RT ± chemotherapy | 224 (67.7) | 66 (56.4) | 158 (73.8) | |
| Chemotherapeutic regimen | | | | |
| Asparaginase–containing | 125 (39.3) | 55 (49.1) | 70 (34.0) | 0.008 |
| Anthracycline–based | 193 (60.7) | 57 (50.9) | 136 (66.0) | |

*(continued on next page)*

**Table 2** (*continued*)

| Parameters[a] | Total *n* (%) | AGR < 1.3 *n* (%) | AGR ≥ 1.3 *n* (%) | *P* value |
|---|---|---|---|---|
| Radiation dose (Gy) | 54.6 (18.0–74.0) | 54.6 (20.0–74.0) | 54.6 (18.0–64.0) | 0.166 |
| Chemotherapy cycles | 4 (0–10) | 4 (1–9) | 4 (0–10) | 0.933 |

**Notes.**
[a]Continuous variables are presented as medians (range), and categorical variables are shown as frequencies and percentages.

AGR, albumin to globulin ration; BMI, body mass index; CRP, C-reactive protein; ECOG, Eastern Cooperative Oncology Group; ESR, erythrocyte sedimentation rate; EUAT, extra-upper aerodigestive tract; IPI, International Prognostic Index; LDH, lactate dehydrogenase; NKPI, natural killer/T-cell lymphoma prognostic index; RT, radiotherapy; UAT, upper aerodigestive tract; WBC, white blood cell.

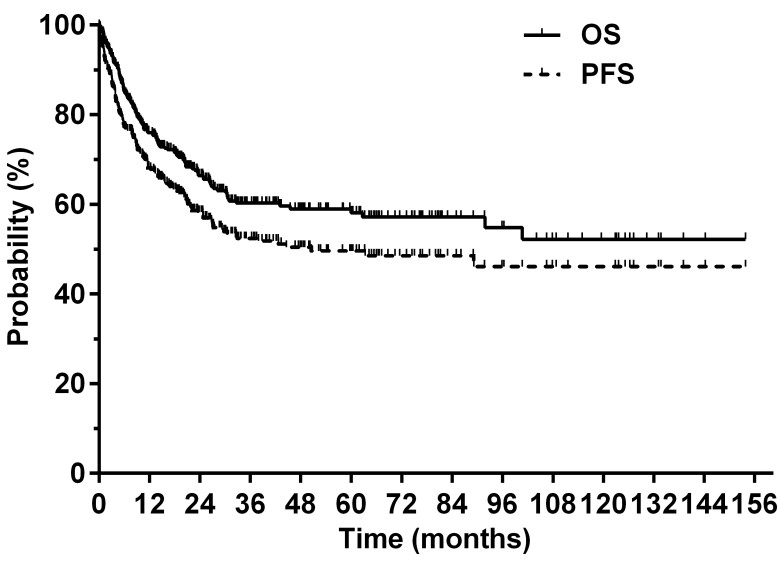

**Figure 1  Overall survival (OS) and progression-free survival (PFS) for the entire cohort.**

## Prognosis and univariate screen of prognostic factors

At a median follow-up of 37.1 months (range, 4.5–153.8 months) for surviving patients, the 5-year OS and PFS for the entire cohort were 58.1 and 49.6%, respectively (Fig. 1). A total of 127 (38.4%) patients died from lymphoma ($n = 119$), treatment-related complications ($n = 3$), comorbidities ($n = 4$), or unknown causes ($n = 1$). Patients with an AGR < 1.3 had a significantly poorer prognosis compared with patients with an AGR ≥ 1.3. The 5-year OS rates were 44.5 and 65.2% in the low and high AGR groups, respectively ($P < 0.001$, Fig. 2A). The 5-year PFS rates were 33.1 and 57.4% in the low and high AGR groups, respectively ($P < 0.001$, Fig. 2B).

Table 3 shows the results from the univariate screening of prognostic factors for OS and PFS. The hazard ratio (HR) of each unit increase in AGR was 0.32 (95% CI [0.17–0.61], $P < 0.001$) for OS. The HR for the OS of the patients in the low AGR group was 2.09 (95% CI [1.47–2.97], $P < 0.001$) compared with the high AGR group. Higher mortality risks were observed in patients aged >60 years, those with an ECOG score ≥ 2, a primary lesion in the EUAT, Ann Arbor stage III-IV, B symptoms, elevated LDH, involvement of regional lymph nodes, and those with two or more extranodal lesions. Each unit increase in CRP,

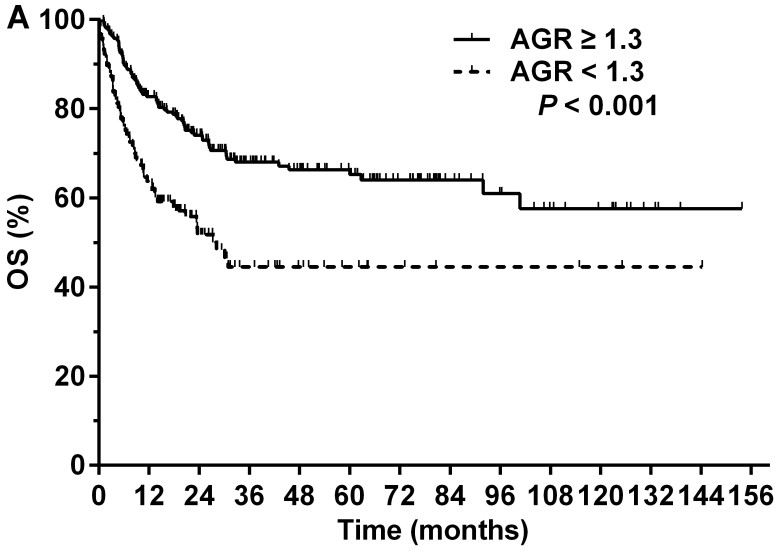

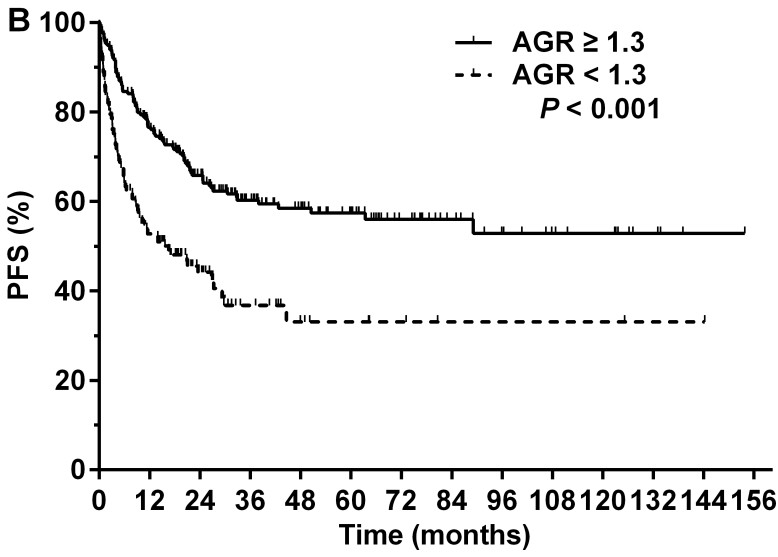

**Figure 2** **Prognosis of patients with NK/T-cell lymphoma according to the pretreatment serum albumin to globulin ratio (AGR).** (A) Overall survival (OS) and (B) progression-free survival (PFS) for patients with NK/T-cell lymphoma with a pretreatment serum AGR $\geq$ 1.3 (solid line) or <1.3 (dashed line).

IPI, or NKPI was associated with an increase in mortality, while each unit increase in serum total protein, albumin, white blood cell (WBC) count, or lymphocyte count was associated with a significant decrease in mortality. In terms of treatment modality, a significantly worse prognosis was observed in patients who received chemotherapy alone (compared with RT with or without chemotherapy) or anthracycline-based chemotherapy (compared with ASP-containing regimens). The BMI, neutrophil count, ESR, and serum creatinine had no significant impact on mortality risk.

**Table 3  Univariate analysis of prognostic factors in patients with NK/T-cell lymphoma.**

| Variable | OS | | PFS | |
|---|---|---|---|---|
| | HR (95% CI)[a] | P value | HR (95% CI)[a] | P value |
| Gender (female vs. male) | 0.70 (0.47–1.05) | 0.081 | 0.80 (0.56–1.14) | 0.215 |
| Age (>60 vs. ≤60 years) | 2.27 (1.51–3.40) | 0.000 | 1.61 (1.07–2.41) | 0.022 |
| BMI | 0.97 (0.92–1.02) | 0.234 | 0.99 (0.95–1.04) | 0.744 |
| ECOG score (≥2 vs. 0–1) | 4.04 (2.57–5.93) | 0.000 | 4.48 (3.14–6.39) | 0.000 |
| Primary site (EUAT vs. UAT) | 1.75 (1.09–2.83) | 0.022 | 2.36 (1.54–3.60) | 0.000 |
| Stage (III–IV vs. I–II) | 3.63 (2.50–5.27) | 0.000 | 3.88 (2.75–5.48) | 0.000 |
| B symptoms (yes vs. no) | 1.65 (1.16–2.36) | 0.006 | 1.77 (1.27–2.45) | 0.001 |
| LDH (elevated vs. normal) | 2.58 (1.81–3.67) | 0.000 | 2.30 (1.67–3.19) | 0.000 |
| Involvement of regional lymph nodes | 2.10 (1.48–2.98) | 0.000 | 2.15 (1.57–2.97) | 0.000 |
| Extranodal sites (≥2 vs. 0–1) | 4.38 (2.94–6.53) | 0.000 | 5.05 (3.46–7.38) | 0.000 |
| AGR per unit increase | 0.32 (0.17–0.61) | 0.000 | 0.38 (0.22–0.68) | 0.001 |
| AGR (<1.3 vs. ≥1.3) | 2.09 (1.47–2.97) | 0.000 | 2.09 (1.51–2.88) | 0.000 |
| Total protein | 0.91 (0.96–0.98) | 0.000 | 0.97 (0.96–0.99) | 0.000 |
| Serum albumin | 0.90 (0.87–0.93) | 0.000 | 0.90 (0.87–0.93) | 0.000 |
| WBC count | 0.92 (0.85–0.99) | 0.026 | 0.94 (0.87–1.00) | 0.062 |
| Neutrophil count | 0.95 (0.88–1.04) | 0.269 | 0.98 (0.91–1.05) | 0.511 |
| Lymphocyte count | 0.50 (0.37–0.68) | 0.000 | 0.52 (0.39–0.69) | 0.000 |
| CRP ($n = 246$) | 1.01 (1.00 –1.02) | 0.008 | 1.01 (1.00–1.02) | 0.003 |
| ESR ($n = 117$) | 1.00 (0.99–1.01) | 0.647 | 1.00 (0.99–1.01) | 0.737 |
| Serum creatinine | 1.00 (0.99–1.01) | 0.552 | 1.00 (0.99–1.01) | 0.979 |
| IPI score | 1.74 (1.55–1.95) | 0.000 | 1.70 (1.53–1.89) | 0.000 |
| NKPI score | 1.65 (1.43–1.90) | 0.000 | 1.65 (1.45–1.88) | 0.000 |
| Treatment (chemo alone vs. RT ± chemo) | 4.55 (3.19–6.49) | 0.000 | 4.31 (3.12–5.95) | 0.000 |
| Chemotherapeutic regimens (anthracycline-based vs. ASP-containing) | 1.92 (1.26–2.93) | 0.002 | 2.00 (1.37–2.92) | 0.000 |

**Notes.**

[a]The hazard ratio gives the increase in risk for each unit increase for the continuous variables and gives the increased risk relative to the reference category for the categorical variables.

AGR, albumin to globulin ration; ASP, asparaginase; BMI, body mass index; CI, confidence interval; CRP, C-reactive protein; ECOG, Eastern Cooperative Oncology Group; ESR, erythrocyte sedimentation rate; EUAT, extra-upper aerodigestive tract; HR, hazard ratio; IPI, International Prognostic Index; LDH, lactate dehydrogenase; NKPI, natural killer/T-cell lymphoma prognostic index; OS, overall survival; PFS, progression free survival; RT, radiotherapy; UAT, upper aerodigestive tract; WBC, white blood cell.

## Multivariate analysis of prognostic factors

The variables that had a significant impact on survival in the univariate analysis were included in the multivariate Cox regression model (Table 4). Serum total protein and albumin were excluded because they were used to calculate AGR. The IPI and NKPI were excluded because they were calculated using several individual factors (i.e., age, ECOG score, stage, B symptoms, LDH, regional lymph node involvement, and extranodal sites ≥ 2) that were already included in the model. CRP was excluded as it is a constituent of serum globulin and affects the calculation of AGR. We ran two separate Cox regression models using AGR as a binary variable (<1.3 vs. ≥1.3) and as a continuous variable (per unit increase). After adjusting for confounding variables, an AGR < 1.3 remained an independent adverse predictor for both OS (HR: 1.74, 95% CI [1.18–2.56], $P = 0.005$) and

**Table 4  Multivariate analysis of prognostic factors in patients with NK/T-cell lymphoma.**

| Variable | OS | | PFS | |
|---|---|---|---|---|
| | HR (95% CI)[a] | P value | HR (95% CI)[a] | P value |
| Age (>60 vs. ≤60 years) | 2.82 (1.80–4.41) | 0.000 | 1.87 (1.23–2.84) | 0.004 |
| ECOG score (≥2 vs. 0–1) | 2.18 (1.29–3.67) | 0.004 | 1.56 (1.02–2.40) | 0.040 |
| Primary site (EUAT vs. UAT) | 2.34 (1.29–4.22) | 0.005 | – | – |
| Stage (III–IV vs. I–II) | – | – | – | – |
| B symptoms (yes vs. no) | – | – | – | – |
| LDH (elevated vs. normal) | 1.54 (1.01–2.36) | 0.047 | – | – |
| Involvement of regional lymph nodes | – | – | 1.56 (1.10–2.20) | 0.013 |
| Extranodal sites (≥2 vs. 0–1) | 2.14 (1.31–3.50) | 0.002 | 2.50 (1.60–3.91) | 0.000 |
| AGR (<1.3 vs. ≥1.3) | 1.74 (1.18–2.56) | 0.005 | 1.73 (1.21–2.48) | 0.003 |
| Lymphocyte count (k/cc) | 0.69 (0.51–0.92) | 0.012 | 0.75 (0.58–0.97) | 0.028 |
| Treatment (chemo alone vs. RT ± chemo) | 2.88 (1.90–4.36) | 0.000 | 2.61 (1.81–3.75) | 0.000 |
| Chemotherapeutic regimens (anthracycline-based vs. ASP-containing) | 2.32 (1.51–3.56) | 0.000 | 2.51 (1.71–3.69) | 0.000 |

**Notes.**

[a]The hazard ratio gives the increase in risk for each unit increase for the continuous variables and gives the increased risk relative to the reference category for the categorical variables.

AGR, albumin to globulin ration; ASP, asparaginase; CI, confidence interval; ECOG, Eastern Cooperative Oncology Group; EUAT, extra-upper aerodigestive tract; HR, hazard ratio; LDH, lactate dehydrogenase; OS, overall survival; PFS, progression free survival; RT, radiotherapy; UAT, upper aerodigestive tract.

PFS (HR: 1.73, 95% CI [1.21–2.48], $P = 0.003$). Similarly, each unit increase in AGR was an independent predictor for improved OS (HR: 0.33, 95% CI [0.17–0.66], $P = 0.002$) and PFS (HR: 0.40, 95% CI [0.22–0.75], $P = 0.004$) in the other model.

### Additional analyses

Further subgroup analyses were performed to determine if the prognostic significance of AGR was merely a function of hypoalbuminemia or early death. Among the 293 patients with normal serum albumin levels (defined as albumin ≥ 35 g/L), an AGR < 1.3 was associated with significantly worse 5-year OS (53.3 vs. 67.7%, $P = 0.005$) and PFS (44.3 vs. 59.6%, $P = 0.005$). Among the 281 patients who survived for more than 6 months after diagnosis, those with an AGR < 1.3 had a significantly worse 5-year OS (58.6 vs. 72.3%, $P = 0.017$) and PFS (44.0 vs. 63.7%, $P = 0.004$).

Subgroup analyses were also performed based on treatment modalities. An AGR < 1.3 significantly worsened the 5-year OS (19.9 vs. 36.6%, $P = 0.013$, Fig. 3A) and PFS (11.6 vs. 31.8%, $P = 0.009$) in the 107 patients who received chemotherapy alone. Among the 224 patients who received RT, those with an AGR < 1.3 had a slightly worse 5-year OS (63.9 vs. 74.6%, $P = 0.070$, Fig. 3B) and significantly inferior 5-year PFS (52.6 vs. 66.4%, $P = 0.037$) compared with those with a higher AGR. In the 193 patients who received anthracycline-based chemotherapy, an AGR < 1.3 was associated with significantly worse 5-year OS (33.0 vs. 58.8%, $P < 0.001$, Fig. 3C) and PFS (21.9 vs. 48.9%, $P < 0.001$). Similarly, an AGR < 1.3 was also associated with significantly inferior 5-year OS (63.5 vs. 82.0%, $P = 0.013$, Fig. 3D) and PFS (51.5 vs. 79.3%, $P = 0.008$) in patients who received ASP-containing chemotherapy.

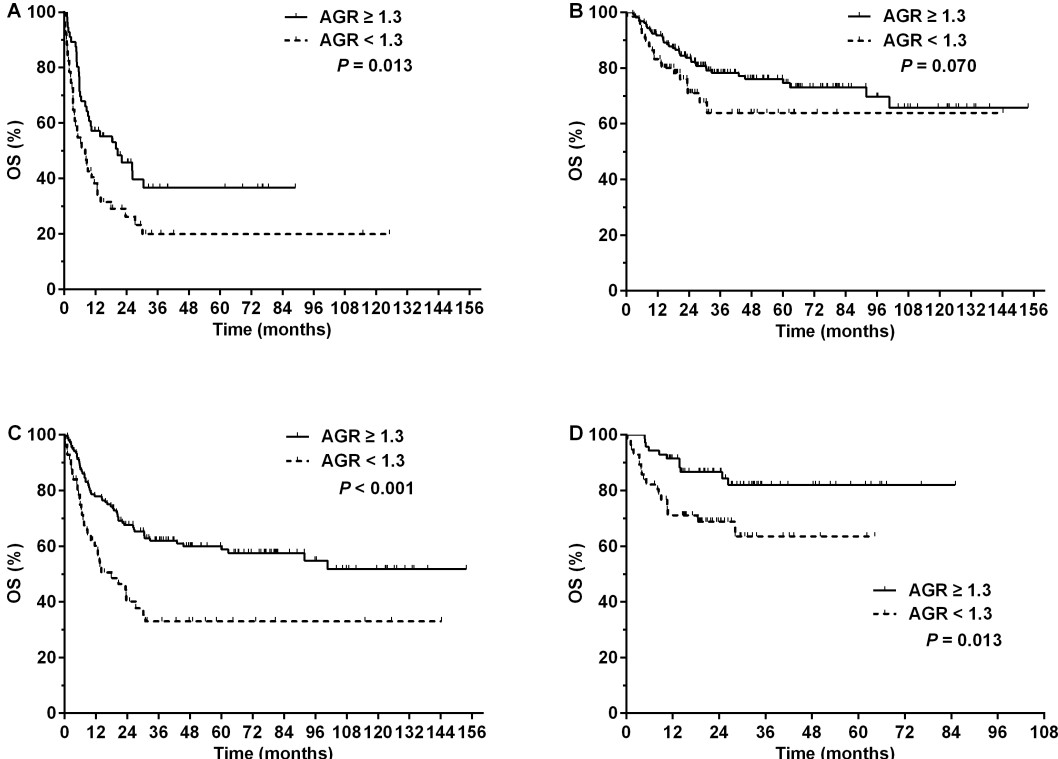

**Figure 3  Overall survival (OS) for patients with NK/T-cell lymphoma with different pretreatment serum albumin to globulin ratios (AGRs) according to treatment modalities.** OS for patients with a pretreatment AGR ≥ 1.3 (solid line) or <1.3 (dashed line) who received (A) chemotherapy alone, (B) radiotherapy with or without chemotherapy, (C) anthracycline-based chemotherapy, or (D) asparaginase-containing chemotherapy.

As shown in Table 2, the IPI stratified patients into low-risk (scored 0–1, 259 cases, 78.2%), intermediate-risk (scored 2–3, 50 cases, 15.1%), and high-risk groups (scored 4–5, 22 cases, 6.6%). These three groups differed substantially in their prognoses (3-year OS: 70.5, 31.0, and 4.5% for the low-, intermediate-, and high-risk groups, respectively, $P < 0.001$, Fig. 4A). However, the IPI low-risk group could be further divided into two separate subgroups with significantly different survival (3-year OS: 75.3 vs. 58.4%, $P = 0.016$) using an AGR < 1.3 or ≥1.3. Accordingly, the entire cohort could be classified into four groups with different survivals using the IPI in combination with AGR (Fig. 4B). AGR failed to distinguish prognostic subsets among patients in the IPI intermediate- or high-risk groups (Fig. S1).

Similarly, the NKPI stratified patients into low-risk (scored 0, 101 cases, 30.5%), intermediate-risk (scored 1–2, 167 cases, 50.5%), and high-risk groups (scored 3–4, 63 cases, 19.0%) with significantly different prognoses (3-year OS: 77.9, 61.6, and 28.1% for the low-, intermediate-, and high-risk groups, respectively; $P < 0.001$, Fig. 4C). AGR differentiated patients with different prognoses in the NKPI high-risk group (3-year OS: 38.9 vs. 20.1%, $P = 0.047$) and divided the entire cohort into four risk groups in combination with the NKPI (Fig. 4D). AGR failed to distinguish prognostic subsets among patients in the NKPI low- or intermediate-risk groups (Fig. S1).

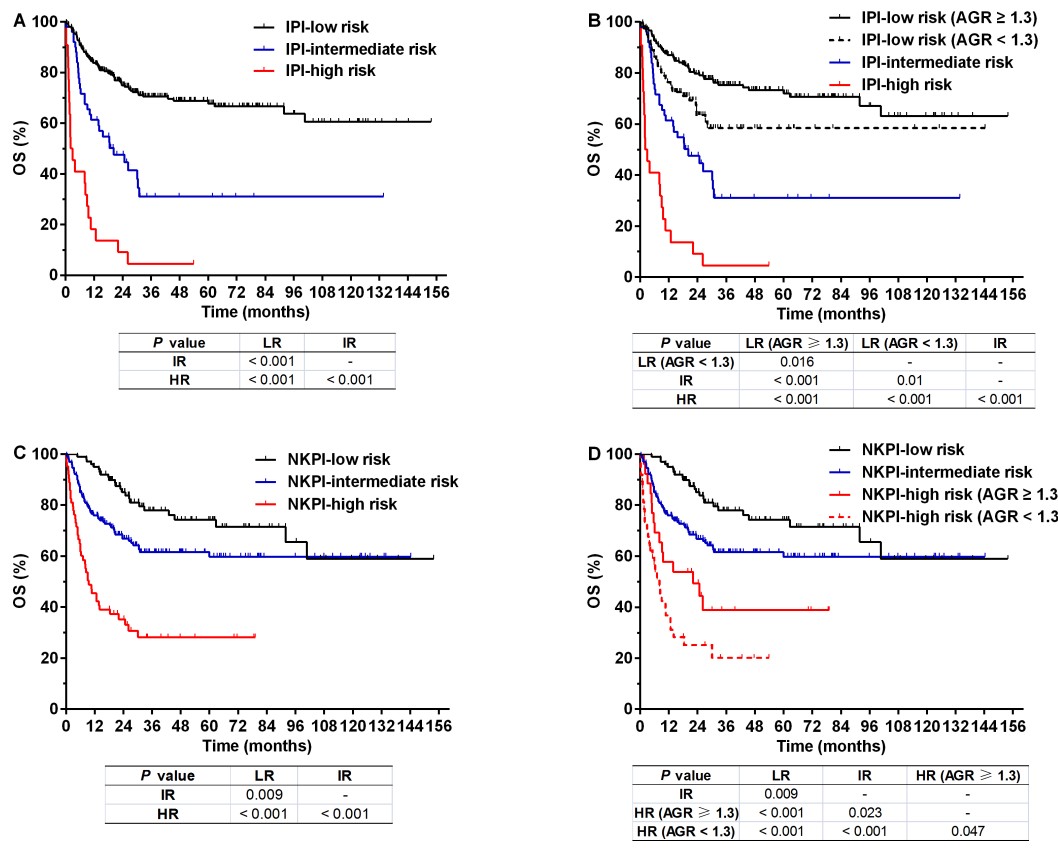

**Figure 4** **Overall survival (OS) for patients with NK/T-cell lymphoma according to prognostic indexes.** OS for patients stratified by (A) the International Prognostic Index (IPI), (B) IPI combined with albumin to globulin ratio (AGR), (C) natural killer/T-cell lymphoma prognostic index (NKPI), and (D) NKPI combined with AGR. *Abbreviations:* LR, low risk; IR, intermediate risk; HR, high risk.

## DISCUSSION

To our knowledge, this is the first study reporting the clinical and prognostic implications of pretreatment AGR in patients with NKTCL. Our findings demonstrated notable differences in clinical behavior between the low and high AGR groups. Patients with a pretreatment AGR < 1.3 presented with significantly more adverse clinical features compared with those with an AGR ≥ 1.3, and a low AGR was independently associated with a poorer prognosis. In addition, AGR could further distinguish patients with different prognoses in the IPI low-risk group and NKPI high-risk group, and AGR in combination with these indices achieved an improved prognostic stratification for the entire cohort.

In previous studies, AGR has been reported as a predictor of mortality in several types of cancer, including breast cancer (*Azab et al., 2013a*), colorectal cancer (*Azab et al., 2013b*; *Shibutani et al., 2015*), lung cancer (*Duran et al., 2014*), and nasopharyngeal carcinoma (*Du et al., 2014*). A recent large-scale study also revealed that low AGR was a risk factor for cancer incidence and mortality in a healthy population (*Suh et al., 2014*). One possible underlying mechanism is an inflammatory reaction, which has been reported to be involved in the tumor's growth, invasion, metastasis, and resistance to treatment (*Balkwill & Mantovani,*

*2001*; *Coussens & Werb, 2002*; *Colotta et al., 2009*). Albumin and globulins are the two primary constituents of serum total protein. Hypoalbuminemia is generally regarded as an indicator of malnutrition. However, many studies have demonstrated that the cancer-related systemic inflammatory response has a more significant impact on hypoalbuminemia (*McMillan et al., 2001*), which is mainly attributed to the cytokine-induced suppression of albumin synthesis and its increased degradation (*Andersson et al., 1990*; *Fearon et al., 1998*). Serum globulins, including immunoglobulins, CRP, complement, and other acute-phase proteins, play a pivotal role in immunity and inflammation (*Gabay & Kushner, 1999*). Both decreased albumin and increased globulin have been reported to be associated with severity and poor outcome for several types of cancer (*Boonpipattanapong & Chewatanakornkul, 2006*; *Siddiqui et al., 2007*; *Gupta & Lis, 2010*; *Guthrie et al., 2013*). Therefore, a low AGR as a combination of these two parameters is considered a more robust predictor of an enhanced inflammatory process and a poorer prognosis.

NKTCL is a relatively rare type of lymphoid malignancy. This disease demonstrates prominent inflammatory characteristics in terms of its pathological, laboratory, and clinical features. Tumor tissues frequently exhibit an angioinvasive pattern with extensive coagulative necrosis and a diffuse polymorphous infiltrate. The typical manifestation involves a destructive nasal lesion with ulceration and purulent discharge. B symptoms (fever, night sweats, and weight loss) are usually observed (52% in our cohort) (*Chan, Jaffe & Ralfkiaer, 2001*; *Chan et al., 2008*). In addition, high levels of inflammatory markers, such as CRP and soluble interleukin-2 receptor (sIL-2R), were found to be predictive of poor survival in NKTCL patients in previous studies (*Li et al., 2013*; *Hanakawa et al., 2014*). In the present study, NKTCL patients with low AGR experienced significant adverse clinical events, including advanced cancer stage, elevated LDH, B symptoms, multiple extranodal involvement, higher IPI and NKPI scores, and an inferior performance status. However, after controlling for these confounding variables, low AGR remained an independent predictor of poorer survival. Further, low AGR maintained its prognostic power in the patients who survived beyond 6 months after their initial diagnosis, suggesting that its predictive value was not merely due to the relatively high short-term mortality observed with this aggressive type of lymphoma.

The exact mechanism underlying the predictive power of AGR in NKTCL remains to be further investigated. Low AGR was associated with lower BMI and serum albumin in our cohort (Table 2). However, BMI had no significant survival impact, and AGR still maintained its predictive value in patients with normal albumin levels (88.5% of the cohort). Conversely, a low AGR correlated with stronger inflammatory clinical and laboratory profiles, including more frequently observed B symptoms and higher ESR and serum CRP levels. Both B symptoms and a high CRP significantly worsened the prognosis in our cohort. Thus, we speculate that chronic inflammation (rather than malnutrition) plays a major role in the prognostic power of AGR. In addition, patients with active inflammatory disorders and liver or kidney disease were excluded before the analysis, negating the possibility that the association between a low AGR and survival was due to the aforementioned conditions.

The IPI has been the most commonly used tool for risk stratification in non-Hodgkin's lymphoma (*Shipp et al., 1993*), but its role in NKTCL remains controversial (*Chim et al., 2004*). The newly proposed NKPI appears to have better prognostic potential than the IPI (*Lee et al., 2006*). However, the factors included in the two models are primarily related to tumor burden (stage, LDH, number of extranodal diseases, and lymph node involvement) and patient characteristics (age and performance status). The only inflammation-related parameter is B symptoms (in the NKPI model). Consistent with previous reports (*Lee et al., 2006*; *Au et al., 2009*), our results revealed that the IPI placed 78.2% of patients in the low-risk group. Moreover, the absolute 3-year survival differences between the IPI low- and intermediate-risk groups (39.5%, Fig. 4A) and between the NKPI intermediate- and high-risk groups (33.5%, Fig. 4C) were relatively large, suggesting imbalanced distributions in these risk groups. After adding AGR into the two models, patients with different outcomes were further discriminated in the IPI low-risk and in the NKPI high-risk groups. Additionally, AGR in combination with the IPI or NKPI divided the entire cohort into four risk groups with significant but more balanced survival differences (Figs. 4B and 4D). Our results suggest that introducing an inflammation-related parameter such as AGR may improve the discriminative power of the current IPI and NKPI, a notion that requires further validation in other cohorts.

Due to the rarity of this disease, no standard treatment regimen has been established for NKTCL. However, it is well accepted that RT is the most important treatment for early-stage NKTCL (*Li et al., 2006*; *Huang et al., 2008*; *Li et al., 2012*; *Bi et al., 2013*). Anthracycline-based chemotherapeutic regimens have demonstrated disappointing efficacy (*Wang et al., 2008*; *Kim et al., 2009*), but regimens containing L-ASP or pegaspargase have shown promising responses based on the results from retrospective studies (*Jaccard et al., 2011*; *Yamaguchi et al., 2011*; *Kwong et al., 2012*). In our cohort, the low AGR group received significantly less RT and more ASP-containing chemotherapy (Table 2), both of which were independent predictors of outcome (Table 4). However, low AGR maintained its adverse impact on survival in patients who underwent RT or no RT, as well as in patients who received anthracycline-based or ASP-containing chemotherapy (Fig. 3). These results suggest that NKTCL patients with low AGR require therapy that is more effective. Anti-inflammatory therapy (e.g., non-steroidal anti-inflammatory agents) has been under investigation for lymphoid malignancies and may be a potential therapeutic choice for NKTCL (*Braun et al., 2012*; *Kumar et al., 2012a*; *Kumar et al., 2012b*). This option requires further exploration.

Several limitations exist in our study due to its retrospective, single-center design. The AGR was only assessed before first-line treatment, and changes in AGR during treatment and follow-up were not routinely evaluated. Therefore, we were unable to analyze the dynamic alterations to AGR and its correlation with treatment response and prognosis. Specific inflammatory and nutritional markers (such as cytokine levels and prealbumin) were not measured, and these assessments might have strengthened the study. In addition, other concurrent disorders that may have impact on the level of globulin, including monoclonal gammopathy of undetermined significance (MGUS), were not routinely assessed, which may have interfered with the interpretation of the results in this study.

## CONCLUSIONS

Low pretreatment AGR is an independent predictor of poor prognosis in patients with NKTCL and provides better risk stratification in combination with the currently used prognostic indexes. As AGR may be measured at low cost through routine pretreatment laboratory tests, it could serve as a simple and effective predictor of prognosis in patients with NKTCL. In addition, the results of this single-center study must be independently validated in other cohorts.

## ACKNOWLEDGEMENTS

We thank all of the physicians at Sun Yat-sen University Cancer Center for allowing us to include their patients.

### Funding

This study was funded by the National-Eleventh Five Technology Major Project (No. 2008ZX09312-002 and 2012ZX09301 to Wen-qi Jiang). The funders had no role in study design, data collection and analysis, decision to publish, or preparation of the manuscript.

### Grant Disclosures

The following grant information was disclosed by the authors:
National-Eleventh Five Technology Major Project: 2008ZX09312-002, 2012ZX09301.

### Competing Interests

The authors declare there are no competing interests.

### Author Contributions

- Xi-wen Bi conceived and designed the experiments, performed the experiments, analyzed the data, wrote the paper, prepared figures and/or tables, collected the clinical and follow-up data of patients.
- Liang Wang and Wen-wen Zhang performed the experiments, analyzed the data, wrote the paper, prepared figures and/or tables, collected the clinical and follow-up data of patients.
- Shu-mei Yan, Peng Sun, Yi Xia and Zhi-ming Li performed the experiments, prepared figures and/or tables.
- Wen-qi Jiang conceived and designed the experiments, reviewed drafts of the paper.

### Human Ethics

The following information was supplied relating to ethical approvals (i.e., approving body and any reference numbers):
Sun Yat-sen University Cancer Center IRB
Approval No. B2015-054-01.

## Data Availability

The raw data was supplied as Table S1.

## Supplemental Information

Supplemental information for this article can be found online at http://dx.doi.org/10.7717/peerj.1742#supplemental-information.

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
