# Peer review of "The pretreatment albumin to globulin ratio predicts survival in patients with natural killer/T-cell lymphoma"

_PeerJ, doi:10.7717/peerj.1742_

## Round 0.1 · original submission · Minor Revisions

I agree with the two reviewers that this paper adds to the literature regarding the prediction of survival in natural killer/T-cell lymphoma patients. I also agree with the reviewers in that I found it a pleasure to read. I had some minor suggestions for improvement which were similar to reviewer two:

1. Please make a clearer rationale for the need for a new survival prediction test in this cohort of patients, particularly in relation to some of the issues with the current tests.
2. Please provide a little bit more detail in the introduction regarding the possible role of globulins and albumin in influencing survival in this patient group.

Reviewer 1 ·

Basic reporting

No comments

Experimental design

Appropriate and thorough.

Validity of the findings

No comments

Additional comments

This manuscript was a pleasure to read. It has been very well written and the work provides considerable benefit to the area of research. I feel it provides excellent data upon which further studies can be undertaken to assess the validity and reliability of using the AGR for survival in patients with natural killer T-cell lymphoma.

Reviewer 2 ·

Basic reporting

This is an interesting study. This reviewer has a few queries, outlined next:

It would appear that the biological rationale for this study is entirely based on sentence beginning line 65. However, the biological relevance of globulins / albumin is not further explained, and the next sentence beginning 'Therefore..' on line 67 is premature; it is not clear to me from the introduction why AGR is relevant to NKTCL.

It also not clear why a new prognostic indicator in NKTCL is needed? It is stated in the intro that other prognostic indicators are controversial, but it is not clear to the reader why these are controversial

Experimental design

Was electrophoresis or immunofixation carried out to assess serum globulin appearance in NKCTL? were patients assessed for other concurrent disorders including MGUS which can increase globulin size by over 10-20g/L and is common in older age.

The authors acknowledge that relevant proteins (e.g pre albulim) were not measured in this study. did the authors measure indices of renal function that could have enabled them to control for renal function in their analyses? It is stated that patients with renal disease were excluded, so there is some acknowledgement that renal function is an important consideration. was creatinine, or better, c-cystatin, or free light chains, measured? Given the broad range of age in this study, renal function may be useful for any prognostic justification of AGR

Validity of the findings

the validity of the findings appears sound, but more information on the 'make-up' of serum proteins in NKTCL would be of interest to the reader. and some justification of why certain relevant proteins were not measured (e.g., renal markers / markers of malignancy) would also help the reader accept the findings of this manuscript.

---

## Round 0.2 · accepted · Accept

We would like to express our congratulations in that you have addressed the comments of both reviewers on the initial version of the manuscript.

Reviewer 2 ·

Basic reporting

The authors have adequately addressed each of my queries / concerns

Experimental design

The authors have adequately addressed each of my queries / concerns

Validity of the findings

The authors have adequately addressed each of my queries / concerns